# Degenerative Cervical Myelopathy: Insights into Its Pathobiology and Molecular Mechanisms

**DOI:** 10.3390/jcm10061214

**Published:** 2021-03-15

**Authors:** Ji Tu, Jose Vargas Castillo, Abhirup Das, Ashish D. Diwan

**Affiliations:** 1Spine Labs, St. George and Sutherland Clinical School, University of New South Wales, Kogarah, NSW 2217, Australia; ji.tu@student.unsw.edu.au (J.T.); A.Diwan@spine-service.org (A.D.D.); 2Spine Service, St. George Hospital, Kogarah, NSW 2217, Australia; fellow@spine-service.org

**Keywords:** degenerative cervical myelopathy (DCM), cervical spondylotic myelopathy (CSM), spinal cord disorder, spinal cord compression, neck pain, blood-spinal cord barrier, microbes

## Abstract

Degenerative cervical myelopathy (DCM), earlier referred to as cervical spondylotic myelopathy (CSM), is the most common and serious neurological disorder in the elderly population caused by chronic progressive compression or irritation of the spinal cord in the neck. The clinical features of DCM include localised neck pain and functional impairment of motor function in the arms, fingers and hands. If left untreated, this can lead to significant and permanent nerve damage including paralysis and death. Despite recent advancements in understanding the DCM pathology, prognosis remains poor and little is known about the molecular mechanisms underlying its pathogenesis. Moreover, there is scant evidence for the best treatment suitable for DCM patients. Decompressive surgery remains the most effective long-term treatment for this pathology, although the decision of when to perform such a procedure remains challenging. Given the fact that the aged population in the world is continuously increasing, DCM is posing a formidable challenge that needs urgent attention. Here, in this comprehensive review, we discuss the current knowledge of DCM pathology, including epidemiology, diagnosis, natural history, pathophysiology, risk factors, molecular features and treatment options. In addition to describing different scoring and classification systems used by clinicians in diagnosing DCM, we also highlight how advanced imaging techniques are being used to study the disease process. Last but not the least, we discuss several molecular underpinnings of DCM aetiology, including the cells involved and the pathways and molecules that are hallmarks of this disease.

## 1. Introduction

Degenerative cervical myelopathy (DCM), also known as cervical spondylotic myelopathy (CSM), is the commonest cause of chronic spinal cord dysfunction worldwide. It is a significant cause of functional disability and leads to a significant ongoing economic burden to those affected by it, their families and their community [1]. DCM is a chronic, primarily non-traumatic and progressive condition. Structures involved in its pathogenesis include the intervertebral discs, vertebral endplates, osteophytes, zygapophyseal and uncovertebral joints and ligaments such as the ligamentum flavum or the posterior longitudinal ligament [2,3]. Although a few papers have been written about its natural history, pathophysiology and treatment, little is known about the molecular mechanisms underlying this condition.

## 2. Epidemiology

The prevalence of DCM in the general population is unknown. A magnetic resonance imaging (MRI) study of asymptomatic individuals showed that up to 25% of the subjects who were less than 40 years old had radiological findings compatible with cervical spondylosis. The incidence of such findings was 60% amongst people older than forty [4]. A cervical disc bulge was present in 88% of 1211 healthy volunteers in another study [5]. With aging, the frequency, size and number of bulging discs increases, while the sagittal diameter and axial area of the dural sac and spinal cord decrease, making this condition a formidable problem in the aging population [6,7].

It has been estimated that degenerative conditions of the spine account for more than 50% of all non-traumatic spinal cord injuries in the United Stated and Japan, and 22% in Australia [8]. The regional incidence for DCM is estimated to be 76 per million in North America, 26 per million in Europe and 6 per million in Australia [8]. This number does not include the patients who may have radiological findings of DCM without symptoms or with very mild symptoms.

The proportion of patients with DCM who underwent surgical treatment was estimated as 1.6 per 100,000 inhabitants [9]. Predicting a patient’s potential for functional recovery before and after surgical decompression remains elusive largely due to the uncertain natural progression of spinal cord pathophysiology [10,11]. This lack of understanding makes the timing and type of treatment offered to patients vary greatly among clinicians.

## 3. Diagnosis

DCM can present clinically as localised neck pain, radiculopathy, myelopathy or a combination of these. Other features of cervical degeneration can include cervicogenic headaches, vertebrobasilar symptoms and precordial pain. All of these makes DCM a part of the differential diagnostic for a diverse number of conditions. This paper focuses on the myelopathy as a consequence of spondylosis; therefore, other conditions like radiculopathy or vertebrobasilar symptoms mentioned prior will not be thoroughly explored, although they are often intertwined with DCM. Despite technological advances, DCM remains a clinical diagnosis [12]. Components needed to make this diagnosis include a history of myelopathic complaints, findings in the physical examination suggestive of myelopathy and this is corroborated by advanced imaging studies showing compression of the spinal cord. However, patients with this condition may have very subtle clinical findings and often, these are not picked up by the unsuspecting clinician.

Diagnosis of DCM is not only difficult due to unsuspecting clinicians, but also because of an overlap in symptoms that may present with other conditions frequently found in the aged population. Regardless, the diagnosis of DCM begins with a thorough history. Clinical symptoms related to DCM include pain or stiffness in the neck, upper extremity clumsiness, gait instability, non-dermatomal numbness or weakness, loss of dexterity, poor coordination, lower extremity weakness, urgency of urination and defecation [13]. Physical examination includes assessment of the cervical spine range of motion (Table 1 and Appendix A) [14]. A limited neck extension should be taken into consideration should surgical treatment be offered, to prevent any iatrogenic hyperextension injury of the neck [15]. Myelopathic signs to be looked for include hyperreflexia, a positive Hoffmann test, a positive Babinski test, clonus and inverted brachioradialis reflex (IBR) [16]. Other possible findings on the physical examination include lower limb spasticity, atrophy of intrinsic hand muscles and corticospinal distribution motor deficits [17]. Furthermore, radiculopathy symptoms can be present as a confounding factor in DCM. A recent study found that over 50% of the patients with DCM had associated radiculopathy [8]. This can complicate the findings in the physical examination, as myelopathy usually presents with hyperreflexia and radiculopathy with hyporeflexia.

**Table 1 jcm-10-01214-t001:** Common findings in the physical examination of patients with degenerative cervical myelopathy (DCM). Each symptom is described separately with a proposed mechanism, as well as their sensitivity and specificity.

Sign/Symptom	Description	Explanation	Sensitivity	Specificity
Hyperreflexia	Reflex greater than 3 on a 0 to 4 scale. (0: absent, 1: hypoactive, 2: normal, 3: hyperactive without clonus, 4: very hyperactive often with clonus.	Interruption of corticospinal and other descending pathways that influence the two-neuron reflex arc due to a suprasegmental lesion. Normally, the cerebral cortex or a number of brainstem nuclei influence the sensory input of the muscle by inhibiting the motor neuron in the anterior horn of the spinal cord. If a descending tract carrying these inhibitory signals is lost, the reflex is augmented.	72%	43%
Hyperreflexia Biceps	Percussion or tapping of the biceps tendon, close to its insertion in the ulna. Greater than 3 on a 0–4 scale.	Mainly C5. Small C6 component.	62%	49%
Hyperreflexia Brachioradialis (BR)	Percussion of the BR tendon distally. Greater than 3 on a 0–4 scale.	Evaluates neurologic integrity of C6.	21%	89%
Hyperreflexia Triceps	Percussion on the distal tendon of the triceps muscle.	Evaluates C7 neurologic integrity.	36%	78%
Hyperreflexia Patella	Percussion on the patellar tendon, with quadriceps relaxed.	Evaluates L4 neurologic integrity.	33%	76%
Hyperreflexia Achilles	Percussion in the Achilles tendon, with a relaxed gastro-soleus muscle.	Evaluates S1 neurologic integrity.	26%	81%
Hoffman	Hand in neutral position, flicking of the distal phalanx of the middle finger causes flexion of the distal phalanx of the thumb and second and third phalanx of the second finger.	Thought to represent a lesion in the corticospinal tracts [18].	59%	84%
Inverted Brachioradialis reflex (IBR)	When eliciting a BR reflex, there is contraction of the finger flexors with diminished BR reflex.	Thought to be caused by a lesion at C5-C6 (damage to the alpha motoneurons) and hyper-active response levels below (C8) [19].	51%	81%
Clonus	Forcefully dorsiflexing the ankle and maintaining pressure on the sole of the foot while observing for rhythmic beats of ankle flexion and extension. More than 3 beats required.	Hyper-active stretch reflexes in clonus are believed to be caused by self- excitation, which is not inhibited by the corticospinal tract (if there is an injury in the spinal cord) [20].	13%	100%
Babinski	Firmly run a pointy instrument, on the lateral part of the sole of the foot, from the heel to the base of the toes. Positive if extension of the Hallux occurs.	The normal response to plantar stimuli is abolished by an upper motor neuron lesion. It is replaced by Babinski’s reflex, where the upward going toe (although anatomically it looks like extension) is part of a flexor reflex, disinhibited by loss of upper motor neurone control, and its receptive field may extend in some instances to the leg or thigh [21].	13%	100%

Patients with DCM more often than not present with positive clinical findings. Seventy nine percent of DCM patients have a positive myelopathic sign and 69% have a positive nerve provocative sign [12]. These numbers are higher in patients with spinal cord changes on an MRI where 95% of patients will have a positive myelopathic sign, especially Hoffmann’s sign (80%). However, patients with cord signal changes can show no signs or symptoms. Close to 20% show no myelopathic sign at the time of presentation and almost 30% lacked hyperreflexia in any reflex arc tested [12]. The absence of these clinical signs should not be a source of doubt for establishing the diagnosis as that may be a cause for delay when offering surgical treatment.

Somatosensory-evoked potentials (SSEPs) and motor evoked potentials (MEPs) are often used to find objective evidence of functional abnormalities of the spinal cord. Often used during surgeries to monitor the well-being of the spinal cord in real time [22], they can also be useful for neurophysiological study for patients with equivocal clinical findings for myelopathy [23]. Some authors have suggested the use of median nerve SSEPs, others, tibial nerve or ulnar nerve SSEPs, and some have found no difference between leg and arms SSEPs.

The predictive value of MEPs and SSEPs for surgical outcomes has not been studied systematically although there are several reports of clinical-electrophysiological correlation. It has been reported that MEPs are more sensitive than SSEPs in detecting chronic myelopathy [24]. SSEPs, however, may have a stronger correlation with surgical outcomes. Due to the anatomical location of the motor pathways and sensory pathways in the spinal cord, the SSEPs usually remain untouched after MEPs may have been affected by anteriorly compressing elements (herniated discs or osteophytes). Once the SSEPs are affected, the potential for a complete recovery after surgery appears to diminish, although this hasn´t been completely proven. Altogether, the role of electrophysiological studies in the diagnosis, follow-up and during treatment for DCM remains to be better defined [25].

The assessment of DCM often includes plain radiographs. Lateral views help evaluate spinal canal narrowing, disc height, the presence of ossification of the posterior longitudinal ligament (OPLL), cervical sagittal alignment and subluxation [26]. Parameters in cervical plain radiographs that are usually measured for assessing DCM are listed in Table 2 and shown in Figure 1A. Patients with DCM often exhibit increased C2–C7 Cobb angles, upper C7 slopes, lower C7 slopes and upper T1 slopes [27].

**Table 2 jcm-10-01214-t002:** Common measurements obtained from standard cervical spine plain radiographs. These measurements are not always performed unless an important sagittal deformity of the spine is deemed responsible for the myelopathy. Some variation exists amongst different authors or according to the position of the patient at the time the radiograph was taken [28].

Radiologic Measures	Normal Values	Explanation
Cobb C1–7/C2–7 angle	18 degrees +/− 12 degrees	The angle between the line parallel to the inferior endplate of C1/C2 to parallel to the inferior endplate of C7.
C7 slope	Normal values vary according to the individual cervical lordosis	Angle between a horizontal line and the superior endplate of C7
T1 slope	Normal values vary according to the individual cervical lordosis	Angle between horizontal plane at T1 endplate
Cervical sagittal vertical alignment (SVA)	15 mm +/− 11 mm	The distance from the posterior, superior corner of C7 to the plumbline from the centroid of C2
Cervical tilt	43 degrees +/− 6 degrees	The angle between two lines, both originating from the centre of the T1 upper end plate; one is vertical to the T1 upper end plate and the other passes through the tip of the dens

A recent report on the correlation between preoperative computed tomography (CT) myelograms and clinical outcomes following surgery showed that patients with greater transverse area of spinal cord at the level of maximum compression had better results [29]. Other investigations such as kinematic CTs have shown limited potential in either demonstrating myelopathy or correlating the findings with clinical outcomes for DCM. CT based investigations have an important role in diagnosing conditions such as OPLL [30].

**Table 3 jcm-10-01214-t003:** Modic type endplate changes represent a classification for vertebral body endplate MRI, first described in 1988 [31]. Often used in the clinical context, these changes are situated in both the body of the vertebrae and in the endplate of the neighbouring disc. It is important to understand that Modic changes do not represent an illness but are a simple descriptive term for radiological findings in MRI.

Modic Type	T1 Findings	T2 Findings	Clinical Correlation
1	Hypointense	Hyperintense	Represent bone marrow oedema and inflammation
2	Hyperintense	Isointense	Conversion of normal hemopoietic bone marrow into fatty marrow as a result of ischemia
3	Hypointense	Hypointense	Represent subchondral bone sclerosis

MRI can provide direct proof of spinal cord compression and should often be the initial investigation; it also plays a role in choosing the right treatment and possibly predicting outcomes. MRI scans allow visualisation of soft tissue structures like intervertebral discs; therefore, early signs of degeneration in them can be detected, as well as in spinal ligaments and other structures not easily seen in other scans. It is unclear whether a direct relationship exists between the quantum of degeneration and cord signal changes independent of canal stenosis.

MRI scans can also detect changes in the signal intensity of the vertebral endplates. When associated with disc degeneration, these are called Modic endplate changes (MECs, Figure 1B) [31]. Three subtypes have been described according to MRI (Table 3). A study found type 2 changes were the most common, especially at the C5–6 and C6–7 levels [32]. However, MECs are a dynamic phenomenon. Mann et al. evaluated the natural course of MECs in 426 patients with neck pain and observed that the prevalence of type 1 MECs increased from 7.4% to 8.2% after 2.5 years follow-up [33]. Similarly, the prevalence of type 2 increased from 14.5% to 22.3%. Twelve segments with type 1 converted to type 2 during the follow-up, while no conversions from type 2 to type 1 were observed.

MRI also offers an opportunity to evaluate spinal canal stenosis (Figure 1C–E). Measuring the anterior–posterior diameter at the region of interest (ROI) is the simplest way that was used in previous studies [34]. However, what is considered a “normal value” for size of the canal varies among individuals. Fehlings et al. developed a method to assess the maximum canal compromise (MCC) after a traumatic cervical spine injury [35]. They evaluated canal size at the ROI by comparing it to the average canal size at the levels above and below it. Although designed for traumatic spinal injury, it has been used for degenerative conditions. Similar to MCC, they also developed the maximum spinal cord compression (MSCC) index to measure the spinal cord compression [36]. Our retrospective study showed that the ratio of the canal diameter to the average of mid-vertebral cephalic and caudal canal diameters is the most sensitive mid-sagittal plane metric for assessing spinal canal stenosis, whereas the ratio of the anteroposterior diameter to the transverse diameter of the cord is the most sensitive axial plane metric [37]. MRI had a role in predicting outcomes in one study: spinal cord atrophy, multilevel T2 hyperintensity, T1 focal hypointensity combined with T2 focal hyperintensity were indicators of poor prognosis for DCM [38].

Certain studies suggest that some spinal cord signal changes can only become evident when a dynamic MRI (flexion/extension MRI) is utilised. A study with 50 patients showed that intensity changes on the spinal cord were made evident with a flexion MRI in 40% of the patients, whereas a neutral MRI only showed these changes in 26% of the patients and an extension MRI only did so in 14% of them [39]. These findings may explain why some MRIs could return negative findings for typical cervical myelopathy, and why these findings might be apparent after surgery in a new MRI. Other authors have reported extension MRIs as helpful to make spinal cord changes evident in patients with DCM, although the relationship between these findings and clinical outcomes is yet to be proven [40].

In addition to conventional MRI, novel techniques have been applied to investigate central nervous system (CNS) pathology, including Diffusion Tensor Imaging (DTI), Diffusion Tensor Tractography (DTT) and Diffusion Basis Spectrum Imaging (DBSI). DTI can estimate the integrity of the tissue microstructure by modelling the diffusion of water within the tissue [41]. DTI is used in brain tumour surgery and has been extrapolated to spinal conditions (Figure 2) [42]. DTI parameters include the Fractional Anisotropy (FA) and Apparent Diffusion Coefficient (ADC). A prospective study found that DTI ratios were more valuable than absolute DTI parameters for the evaluation of DCM, as the latter can be confounded by age and cervical level [43]. DTT is a functional imaging technique, which allows tracking of the nerve fibres based on their FA values and can be demonstrated when the nerve fibres get distorted, disoriented or even interrupted as the severity of the spinal compression varies. DTT and DTI are more valuable than routine MRI scans for diagnosis and predicting outcomes in DCM patients [44,45]. DBSI allows for the quantification of axonal injury, demyelination and inflammation in DCM patients.

Several score systems have been used throughout the years to study DCM. Based mostly on signs or symptoms, their importance relies on the prognostic value they may have and to facilitate comparison of different treatment methods. An overview of the most common ones is detailed in Table 4, including their advantages and shortcomings [46,47].

**Table 4 jcm-10-01214-t004:** Clinicians use scoring systems to categorise the severity of different conditions. Often different classifications arise as different groups come up with their own systems; however, international consensus groups usually choose one system to standardise publications and treatments across the board. This has not been the case with DCM. Several different systems are still been used by different authors based on their preference. The following are the most common classification systems currently in use, along with a guide to their score meaning, presence of radiologic features, short-comings and advantages. Showcasing the complete classifications is beyond the scope of this review. To obtain the complete scoring systems, please follow the link to the reference [15,47,48,49].

Name	Scoring Method	Radiologic Findings	Correlation to Symptoms	Limitations	Advantages
Nurick	0–5. The higher the grade, the more severe the deficit.	No	Affected by gait function (++), lower limbs paresis and paraesthesia and vegetative symptoms (+).	Less accurate post-op scoring; Does not pick up upper extremity disfunction	Evaluates economic situation in connection to gait function.
mJOA	0–17. The lower the score, the more severe the deficits. Normal: 16–17, grade 1: 12–15, grade 2: 8–11, grade 3: 0–7. Upper extremity 23.5%; lower extremity 23.5%; sensory 35.4%; bladder and bowel 17.6%	No	Affected by paraesthesia of lower limbs and paresis of upper limbs (++) and dysdiadochokinesia and vegetative symptoms (+).	Does not take economic factors into consideration	Good for assessing outcomes (post-intervention).
CMS	Upper and lower extremity are analysed separately. 0–5 each. The higher the grade, the more severe the deficit.	Weak correlation between low severity in the lower limb score and C-Spine mal-alignment	Affected by dysdiadochokinesia, gait function and paresis of upper extremity (++) and vegetative symptoms (+)	Does not take economic factors into consideration	Good for assessing function/symptoms of upper/lower extremities/as it evaluates them individually. Good at assessing clinical state and grade of severity of CSM.
EMS	5–18. The lower the score the more severe the deficits. Normal function: 17+, grade 1: 13–16, grade 2: 9–12, grade 3: 5–8. Upper extremity 27.8%, lower extremity 22.2%, coordination 16.7%, paraesthesia/pain 16.0%, bladder and bowel function 16.7%	No	Affected by dysdiadochokinesia (++) and paresis of the upper extremity and vegetative symptoms (+)		Good at assessing clinical state and grade of severity of CSM. Better sensitivity to reveal functional deficit (by assessing proprioception/coordination).
Prolo scale	2–10. The lower the score the more severe the deficits. Normal function: 9+, grade 1: 7 + 8, grade 2: 5 + 6, grade 3: 2–4. Economic status 50%; functional status 50%.	No	Mildly affected by vegetative symptoms (+)	Does not reflect clinical symptoms significantly -Not good for pre-op assessing the grade of severity	Good correlation between high pre-op scores and better outcomes. Good for assessing normalisation¨ and rehabilitation (regained ability for work or for leisure time).

mJOA: modified Japanese Orthopaedic Association; CMS: Cervical Myelopathy Scale; EMS: European Myelopathy Scale.

## 4. Natural History

The natural history of DCM is still not clear. While the nature of the injury and ultimate consequences share similarities with acute spinal cord injuries, the pathophysiology differs [10]. An old descriptive study from 1956 described the average age for the appearance of symptoms to be at around 50 years of age and 70% of the patients were between 40 to 59 years. Out of 120 patients with DCM, 5% of them had a rapid onset of symptoms followed by long periods of remission, 20% had a slow progressive worsening of neurofunction and 75% had a stepwise decline of neurofunction [50]. The progression of symptoms in patients with DCM has been studied. In 1963, a retrospective study of DCM to understand its natural history, found that a majority of patients had a poor prognosis, with more than 87% progressing to moderate or severe disability at the last follow-up [51].

A prospective research in 199 asymptomatic patients with cervical spinal cord encroachment detected by radiology was conducted to find out the effects of traumatic episodes (head, spine, trunk or shoulders) on these patients. A total of 14 episodes were recorded during a median 44 months follow-up, and only one patient developed myelopathy. Meanwhile, 44 patients without a history of trauma developed myelopathic symptoms. It can be inferred that the risk of developing myelopathy in asymptomatic patients with cervical spinal cord encroachment after minor trauma is low [52]. However, another study in patients with OPLL showed that minor trauma is of importance in the development or deterioration of myelopathy in said patients [53].

## 5. Pathophysiology

### 5.1. Spinal Cord Compression and Ischemic Injury

Mechanical compression is the corner stone of spinal cord dysfunction in DCM. Studies on bovine cervical spinal cords showed a different stress distribution between white and grey matter, which varied with strain rate, compression volume and the position of compression. These differences may explain the diverse signs and symptoms found in DCM [54]. In an animal model of chronically compressed spinal cord (tiptoe-walking Yoshimura (twy) mice), *p62* and autophagy markers (autolysosomes and autophagic vesicles) were found to accumulate in neurons, axons, astrocytes and oligodendrocytes. These molecules are linked to neuronal cell death [55]. Fas-mediated apoptosis of neurons and oligodendrocytes and an increase in inflammatory cells were also observed in twy mice and post-mortem human spinal cords samples of DCM patients in a different study [56]. Mechanical compression can also lead to ischemia and hypoxia, which would result in spinal cord dysfunction, similar to that found in acute traumatic spinal cord injuries. The compression can be caused by static and/or dynamic factors. The static factors refer to structural spondylotic abnormalities such as disc degeneration, which result in cervical canal stenosis. The dynamic factors include changes to the normal cervical spine biomechanics and tensile stresses transmitted to the spinal cord from the dentate ligaments, which attach the lateral pia to the lateral dura [57,58].

Ischemic injury was first described in the pathophysiology of degenerative spondylitis in 1948 [59]. Further studies confirmed the observation with human and animal evidence. Ischemia related tissue changes, including flattening of the cord, swelling of myelin and axons, demyelination in the posterolateral and anterolateral columns and neuronal loss in the anterior horns have been observed in the spinal cord of DCM patients. The chronic compression can obstruct branches of the anterior spinal artery with the ensuing ischemic damage, as shown in a series of post-mortem case reports [60]. Researchers found motor disturbances were worsened by induced exacerbated spinal cord hypoperfusion. They proved this by exsanguination plus ligation of the carotid and vertebral arteries in a cervical chronic compression dog model [61,62]. Rodent experiments have also proved that chronic compression of the cervical spinal cord leads to architectural changes of the microvessel network and altered distribution of spinal cord blood flow [63].

### 5.2. Spine Deformity and Instability

Cervical sagittal malalignment is a contributing factor to DCM [27]. The cervical spine has a lordotic disposition, that can be first seen as early as the 9th week of gestation [14]. As mentioned earlier, with aging and the ensuing degeneration, several alignment abnormalities may arise, such as increased lordosis, scoliosis and kyphosis. These changes can compromise the volume of the vertebral canal, reducing the space available for the spinal cord. A study conducted in North America showed moderate negative correlation between cord cross-sectional area and modified Japanese Orthopaedic Association (mJOA) scores in patients with kyphotic deformities in the cervical spine [64]. Kyphotic deformities may lead to spinal cord tethering and stretching, resulting in increased intramedullary pressure and impaired microcirculation, leading to demyelination, neuronal loss and myelopathy [65]. Atlantoaxial joint instability is also believed to be associated with subaxial cervical instability and the appearance of DCM [66,67].

### 5.3. Ossification of the Ligaments

Ossification of the posterior longitudinal ligament (OPLL), anterior longitudinal ligament (OALL) and/or of the ligamentum flavum (OLF) can affect the space available for the spinal cord and subsequently cause DCM [68]. Its incidence in the Japanese population is estimated as between 1.9 and 4.3%, averaging 3.0% in other Asian countries [69]. However, it’s only 0.1 to 1.7% among Caucasian cohorts [70]. Although the mechanism of OPLL remains poorly understood, it shares similarities with diffuse idiopathic skeletal hyperostosis (DISH). Some systemic hormones are considered to play a role in the initiation and development of OPLL, such as 1,25-dihydroxyvitamin D, parathyroid hormone, insulin and leptin, as well as local growth factors, such as transforming growth factor-β (TGF-β) and bone morphogenetic protein (BMP) [71].

### 5.4. Biomechanical Changes

The increased association of DCM with aging raises the issue whether anchoring of the cervical spinal cord by dentate ligaments provides tensile friction to cause microtrauma of the spinal cord, or whether the changing stiffness of the neural tissue and extracellular matrix (ECM) in the spinal cord can possibly make the spinal cord stiffer and susceptible to repetitive micro-injury with progressive age. Such biomechanical non-compressive mechanisms have been explored. Finite element analysis (FEA) showed that intramedullary stress contributes to DCM pathogenesis [54,72]. One study has indicated that a threshold of intramedullary stress to present symptoms of myelopathy actually existed and is related to neurological dysfunction [73]. A 3D finite element model showed that cervical flexion-induced spinal cord stress results in muscle atrophy and weakness [74].

## 6. Risk Factors

### 6.1. Aging

Aging is associated with tissue degeneration and a change in the chemical properties of tissues. Not surprisingly, the prevalence of cervical cord compression increases with increasing age [75]. DCM is uncommon in patients under 40 years of age. Most patients are diagnosed with DCM in their fifth decade of life [76]. A prospective longitudinal study in healthy volunteers revealed that the incidence of foraminal stenosis, posterior disc protrusion and disc space narrowing in MRI was higher in elderly subjects [77]. Aging is also associated with changes in the sagittal alignment of the cervical spine, namely, loss of the physiologic lordosis [78].

### 6.2. Genetic Polymorphism

It has long been speculated that DCM has genetic predisposition [79]. In 2012, a retrospective study based on over 2 million Utah residents showed a relative risk of 5.21 and 1.95 for first degree and third degree DCM patients’ relatives, respectively [80]. Polymorphisms in a number of genes that have been identified as contributing to the development of DCM are listed in Table 5.

**Table 5 jcm-10-01214-t005:** List of genes associated with DCM pathology.

Gene	DCM Features	Reference
Brain-derived neurotrophic factor (BDNF)	Worse mJOA and Nurick scores	[81]
Osteoprotegerin (OPG)	Worse mJOA score	[82]
Osteopontin (OPN)	Worse mJOA score	[83]
Hypoxia inducible factor-1α (HIF-1α)	Worse mJOA score	[84]
Apolipoprotein E (APOE)	Worse mJOA score	[85,86]
BMPs (BMP4, BMP9, BMPR1A)	Radiographic severity of DCM	[87,88]
RUNX2	Responsible for OPLL	[89]
BMP2	Responsible for OPLL	[90]
Vitamin D receptor (VDR)	Radiologic changes and mJOA scores	[89,91]
Vitamin D binding protein (VDBP)	Radiologic changes and mJOA scores	[91]
Collagen IX	Radiologic changes and mJOA scores	[92]
Collagen α2(XI)	Radiographic severity of DCM	[93]

### 6.3. Microbes

One of the emerging risk factors for DCM that has been coming to the fore recently is bacterial infection. Low virulence bacterial infections have been observed in degenerate cervical discs of DCM patients undergoing surgery; however, it is not yet clear if these infections play a role in the development of clinical symptoms [94,95]. *Propionibacterium acnes* and coagulase-negative *Staphylococci* were the most commonly identified bacteria. Interestingly, a recent study indicated that the lumbar intervertebral discs harbour their own unique bacterial population (disc microbiome), and alterations in bacterial diversity (dysbiosis), both in the disc and gut, strongly correlate with disc disorders in back pain patients [96]. Further study is warranted to verify if similar disc microbiome exists in the cervical disc and whether dysbiosis plays any role in DCM pathogenesis and surgery outcomes.

## 7. Molecular Features

### 7.1. Cervical Intervertebral Disc Degeneration

Intervertebral disc (IVD) degeneration is a common finding; 98% of healthy adults show IVD degeneration in their 20s [97]. It is pivotal for the development of cervical spondylosis. The IVD consists of three specialised tissues: the central nucleus pulposus (NP), the outer fibrillar annulus fibrosus (AF) and the cartilage end plates (CEP) that anchor the disc to the adjacent vertebral bones. Most of the molecular studies of IVD degeneration focus on lumbar IVDs, and while it is true that they share similar biologic characteristics, there are several differences between cervical and lumbar IVDs. In human, collagen content is highest in cervical IVD, whereas polyanion concentration is highest in lumbar discs [98]. Compared to the lumbar AF, the fibres of the cervical AF are more perpendicular to the endplates in orientation [99].

IVD degeneration leads to an increased biomechanical stress on the rest of the cervical spine (Figure 3A). It has been shown to increase the shear stress on the vertebral cortical bone which leads to remodelling of this bone and to the formation of osteophytes. These abnormal bony formations can cause DCM and radiculopathy [100,101]. An in vivo study showed that neurotrophins, BDNF and Nerve Growth Factor (NGF) are increased in painful cervical discs and correlated with clinical findings [102]. Revascularisation into the disc is also a feature in DCM [103]. Disrupted disc microenvironment and senescence of IVD cells induce the imbalance between their ECM anabolism and catabolism. The degradation of ECM components and deterioration of the major structural proteoglycan aggrecan result in reduced hydration, loss of disc height and an overall inability to absorb compressive load [104]. During this process, inflammation, cell apoptosis and mitochondrial dysfunction are widely prevalent [105,106].

### 7.2. Blood-Spinal Cord Barrier Dysfunction

The local environment around the blood–spinal cord barrier (BSCB) undergoes profound biochemical and cellular changes with DCM (Figure 3B). The different pathways and interactions involved in this process are not quite completely understood. BSCB is the continuation of the blood–brain barrier (BBB); however, a few morphological and functional differences exist between them [107]. BSCB provides a special immune-privileged environment to the spinal cord, protecting the CNS from neurotoxic insults. These insults may include peripheral immune cell invasion, cytokines and reactive oxygen species (ROS). The presence of these elements leads to neuroinflammation and neurodegeneration [108]. There is evidence that spinal cord trauma leads to dysfunction of the BSCB [107]. Three markers of different size (fluorescently labelled hydrazide, fluorescently labelled bovine serum albumin and immunohistochemically labelled red blood cells) showed greater concentrations in the grey matter than in white matter, and correlated better to the rate of spinal cord compression than to the depth of compression [109]. Longitudinal dynamic contrast-enhanced MRI (DCE-MRI) studies revealed that the BSCB remained compromised even 56 days after moderately severe injury to the spinal cord in an animal model. A significant correlation between decreased BSCB permeability and improved motor recovery was also observed [110].

Endothelial cells are responsible for the integrity of the BSCB. Quantitative loss and dysfunction of these cells can induce impairments in the BSCB, resulting in spinal cord oedema and inflammation [107]. Oestrogens are thought to have an effect on the overall health of the structure, as it has been shown that tamoxifen, an oestrogen-receptor inhibitor, bolsters the BSCB, by means of decreasing tissue oedema and IL-1β production and decreasing myelin loss in spinal cord injury (SCI) [111]. A prospective non-randomised controlled study revealed increased BSCB permeability in DCM patients, as evident from the increased levels on Albumin Q, IgG, and IgA into intrathecal space [112]. The severity of BSCB disruption and the diffusion of IgG were also found to be related to the clinical status. Swelling of the spinal cord can also be seen after BSCB disruption, and it has been found in roughly 8% of patients with DCM [113]. Radiologically, a disruption of BSCB can be seen in the form of positive intramedullary Gadolinium enhancement around the white matter vessels in an MRI sequence [114].

### 7.3. Axonal Injury

An important feature of DCM, axonal injury (Figure 3C), can be evaluated using FA obtained from DTI MRI. The concept underpinning this technology is that water molecules diffuse differently along the tissues depending on the type of tissue, their integrity, architecture and presence of barriers, providing information about its orientation and quantitative anisotropy. Analyses of the FA values of different neural elements provide information about the relative indemnity of said structure. Differences in the FA ratios of DCM patients from different mJOA score subgroups were observed in a recent study [44]. This could mean that the severity of DCM is related to axonal integrity. Decompression of spinal cord was also found to correlate with axonal sprouting in another imaging study, although the clinical implications are not clear [115]. Axonal degeneration can be activated by different stimuli including mechanical injury, axonal transport defects or drugs [116]. Some studies indicate that axonal degeneration may be an early event in neurodegenerative diseases and may precede any radiological findings of compression [117,118]. This observation suggests that there may be other catalysts for axonal injury, besides the aforementioned. The presence of microbial and/or inflammatory metabolites, or potentially micro-trauma, could be one or more of them.

### 7.4. Astrocytes

In 1895, Michael von Lenhossék used the word astrocyte to describe the star-shaped glial cells in vertebrates. They are the most abundant, constituting nearly 1/3 of the cells in the human CNS. Astrocytes perform many important functions in the CNS. They are involved in maintaining homeostasis at the synapse and regulating neuronal signalling. They act as an essential part of BSCB, protecting neurons from oxidative damage by controlling the access of peripheral cells to the spinal cord. They also take part in forming the glial scar after an injury, along with microglia/macrophages and ECM molecules [119]. Astrocytes increase their number and migrate to the damaged site. In severe injuries, they surround the SCI lesions and form a glial scar, acting as a physical barrier to contain the injured area [120,121].

Astrocytes alter the composition of the ECM following an injury. Several ECM components like chondroitin sulphate proteoglycans and tenascins are markedly upregulated in astrocytes after being stimulated [122]. Astrogliosis is the proliferation and hypertrophy of astrocytes, resulting in scar formation via the activation of signalling pathways such as STAT3 and TGF-β. A histological study of horses with chronic compressive myelopathy found astrogliosis a prominent and persistent finding in their spinal cords [123]. Researchers have demonstrated that chronic mechanical compression of the cervical spinal cord leads to astrogliosis in the dorsal horns of the spinal cord [124]. Activated astrocytes express intermediate pro-inflammatory filaments in their membrane, such as glial fibrillary acidic proteins (GFAP), nestin and vimentin. In a rabbit model of unilateral spinal cord compression, the density of GFAP-positive astrocytes was significantly increased, providing evidence they play a role in compressive pathology of the spinal cord [125].

Reactive astrocytes also contribute to the release of both pro- and anti-inflammatory cytokines such as interleukins (IL-1 and IL-6), TGF-β, interferon γ (IFN-γ) and tumour necrosis factor-α (TNF-α). These cytokines modulate inflammation and play a role in secondary injury mechanisms [126]. The release of the chemokine CXCL1 from astrocytes and the subsequent activation of its CXCR2 receptor on neurons is evidence of the crosstalk between the two cell types (Figure 3C). This particular interaction induces descending neuron degeneration in spinal cord [127]. Astrocytes are also involved in neuropathic pain modulation and processing. Toll-like receptor 4 (TLR-4) pathway contributes to astrocyte activation and astrogliosis during chronic pain sensitization in the spinal cord [128]. Animal experiments proved cervical contusion-induced neuropathic pain is associated with persistent astrocyte activation in the superficial dorsal horn [129].

### 7.5. Microglia and Neutrophils

As the resident macrophage cells, microglia are central players in the innate immune response following injury to the CNS (Figure 3C). Under normal circumstances, they patrol their micro-environment in search for abnormal epitopes to trigger a defence response. However, after an injury, they take part in the production of harmful ROS and pro-inflammatory cytokines. They also contribute to the glial scar found around damaged tissue in the CNS [130]. Neutrophils and activated microglia appear in the first few days of SCI and are loaded with destructive oxidative and proteolytic enzymes. Oxidative activity related to myeloperoxidase (MPO) and nicotinamide adenine dinucleotide phosphate oxidase (NADPH oxidase) released by neutrophils are mainly associated with neutrophils and activated microglia, while phagocytic macrophages have weak or no enzyme expression. Matrix metalloproteinase (MMP) 9 is only expressed by neutrophils and is a strong pro-inflammatory molecule [131]. Neutrophils are only detectable for up to ten days after the initial injury, with activated microglia, a few monocytes/macrophages and numerous phagocytic macrophages lingering for weeks to months afterwards.

The main biochemical difference between SCI and DCM is that the latter, being a chronic process, is driven by chronic inflammation, and thus, the molecular markers and characteristic cell types are different to those seen in acute responses. It has been shown that activated macrophages/microglia are the predominant cell types in both the early and late phases of DCM [56]. A chemokine often involved in the chemotaxis of monocytes and leucocytes called fractalkine (CX3CL1) was found to be widely expressed in the membrane of neurons, while its receptor (CX3CR1) is highly expressed on microglia [132]. Animal experiments on ischemic mice shed some light on the role of CX3CR1 during ischemia in the CNS [133,134]. Under ischemic conditions (common in DCM), the development of activated microglia in CX3CR1 knockout mice was significantly impaired. Post-mortem immunohistochemistry revealed CX3CR1 depletion led to a decrease in the activation of microglia/macrophages, while leukocyte recruitment increased. This suggests that CX3CR1 plays a role in the regulation of microglia and neuroinflammation in conditions like DCM [134,135].

Microglia has also been involved in mechanisms for neuropathic pain. It has been shown that inhibiting the function or expression of microglial-produced molecules, such as activated protein-kinases, p38 and other extracellular signal-regulated protein kinase, suppresses the abnormal excitability of dorsal horn neurons found in neuropathic pain [136,137].

### 7.6. Oligodendrocytes

Oligodendrocytes (OLG) support and insulate the axons of neurons (Figure 3C). Abnormalities in OLG are associated with neurological symptoms and are a common finding in acute and chronic spinal cord injuries. An immuno-histochemical study of patients with DCM showed that the distribution of apoptotic OLG was analogous to the degeneration of the long tracts in cervical spinal cord [133]. The relatively low reduced glutathione and high iron concentration in OLG renders them vulnerable to oxidative stress (present in inflammatory conditions of the spinal cord) [138]. The pro-inflammatory cytokines, IL-1β and TNF-α, were found to inhibit the expression of myelin genes in human OLG through the alteration of the cellular redox system [108].

The dysfunction of OLG is deeply related to demyelination. Demyelinated corticospinal tracts are a constant finding in DCM [139,140,141]. However, whether primary demyelination appears as a result of damage to OLG or myelin loss comes secondary to axonal degeneration remains unclear. Demyelination has been identified in compressed spinal cord samples [142,143] and successfully reproduced using toxin-induced models, virus-induced and autoimmune models [144]. This explains the myriad of causes that may lead to this condition. Evidence shows that neuronal and OLG apoptosis contribute to demyelination and Wallerian degeneration, resulting in neurological deficit [145,146]. Decreased myelin content in the spinal cord was shown to be associated with impaired spinal cord conduction [147]. A study using surgery-induced spinal cord compression in a horse model showed that OLG apoptosis immediately occurred after the injury and was consistent with the extent of demyelination. This indicates that OLG apoptosis induced by compression contributes to demyelination [142]. At least two different pathways have been proposed to explain the apoptosis of OLG in DCM: (1) via Endoplasmic Reticulum (ER)–mitochondria interaction (increased caspase-12 and cytochrome *c*) and (2) upregulation of E1F2 (a pro-apoptotic transcription factor associated with the p53 protein in its apoptotic pathway) (Figure 4) [148]. Between ER and mitochondria, mitochondrial fission protein Fission 1 homologue (Fis1) and Bap31 at the ER can combine to form Fis1-Bap31 complex (ARCosome), serving as a platform for caspase-8 activation, leading to apoptosis [149]. E1F2 phosphorylation can enhance CHOP translation, leading to inflammasome activation and cytokines release [150,151].

Fas ligand mediated OLG apoptosis has been shown to contribute to cell death and inflammation in a model of DCM [56]. TNF-α is also a known inducer of apoptosis of neurons and OLG. In the early phases of SCI, TNF-α serves as an external signal triggering apoptosis in OLG, but its role has not been determined in DCM [152]. Apoptosis signal-regulating kinase 1 (ASK1), Jun N-terminal kinase (JNK) and p38 signal pathways were found to be activated in OLG in an animal model of chronic spinal cord compression [153]. Notably, ASK1 can be activated by TNF-α or Fas and act as a mediator of JNK activation. Some counterbalances have also been seen in the spinal cord against apoptotic cascades. Insulin-like growth factor-1 (IGF-1) can protect myelin and oligodendrocytes from TNF-α induced apoptosis [154].

Inflammasome, a cytosolic multiprotein oligomer of the innate immune system responsible for the activation of inflammatory responses, has been detected during inflammatory states in multiple cell types of the CNS, including OLG [155]. Increased intracellular calcium (Ca^2+^) leads to the release of ROS and NLRP3 inflammasome complex activation, which itself facilitates caspase-1 autoactivation and the subsequent proteolytic cleavage and release of IL-1β [156].

A key regulator, phosphatase and tensin homolog (PTEN), regulates Ca^2+^ release from the ER. PTEN can counteract the inositol-1,4,5-trisphosphate receptors (IP3Rs)-induced Ca^2+^ release mediated by AKT phosphorylation [157]. Other chaperone proteins, like sigma-1 receptor (SIG1R)/GRP78, GRP75, fragile histidine triad diadenosine triphosphatase (FHIT) and protein kinase R (PKR)-like endoplasmic reticulum kinase (PERK) also participate in regulating the Ca^2+^ movement between cell members [158]. PTEN is considered to be a major negative regulator of neuronal regeneration in SCI [159]. The role of PTEN in DCM is still elusive, although studies in chronic demyelinating diseases show that PTEN is required during OLG development and repair and its inactivation may lead to loss of myelin and axon integrity [160].

### 7.7. Brain Reorganization

DCM not only displays an array of changes in the cervical spine, but also in the brain. Cortical and cerebellar abnormalities have been found in DCM patient [161,162,163]. The relationship between DCM and brain reorganisation has been shown by blood oxygenation level dependent functional MRI (fMRI) analysis. A study analysing changes in the volume of activation (VOA) between patients with DCM and healthy controls showed changes in VOA are associated with neurological status and can change after surgical decompression [164]. Metabolic profiles in brains were measured by proton MR spectroscopy in 21 DCM patients and 16 healthy volunteers and metabolite levels in the cerebellum were found to be significantly different between these cohorts (Figure 3D). Some of these metabolites, myo-inositol and choline across primary motor cortices, N-acetylaspartate (NAA; marker of neuronal integrity) and glutamate–glutamine in the left motor cortex, and myo-inositol and glutamate–glutamine in the cerebellum, were found to be significantly associated with postoperative clinical status [165]. These metabolic profile changes may arise due to brain reorganization in DCM.

## 8. Treatment

There is a lack of evidence to support a best treatment for patients with DCM. Often, the therapeutic options offered to a patient, regardless if they are surgical or not, depend more on their doctor’s preference instead of strong scientific evidence to support one or another approach [166].

### 8.1. Non-Surgical Treatment

Classic papers described a poor prognosis for DCM (regardless of the type of treatment applied), and thus recommended a non-operative approach [51]. This includes physical therapy with strengthening of the muscles in the neck, back and pelvic girdle to improve gait and pain. Exercises aimed at improving proprioception and balance take a central place when it comes to non-operative measures to assist patients with this condition. Other methods such as heat packs and acupuncture are often used to alleviate the symptoms [51]. A 10-year prospective randomised study found there was no significant difference in outcomes or survival between a conservative and an operative treatment in patients with mild and moderate DCM [166]. A recent systematic review found lack of sufficient evidence to adequately assess the role of non-operative treatment in DCM and a clinically significant gain of function was not observed in the majority of patients following a structured non-operative treatment program [167].

In recent years, neuroactive drugs have shown a potential value for the treatment of DCM. Oestrogens have been found to inhibit glutamate induced apoptosis, by suppressing caspase-3 in neuronal cells [168]. However, some studies showed that tamoxifen, an oestrogen-receptor blocker, can inhibit ROS and lipid peroxidation after ischemia/hypoxia and has been used to treat SCI [169,170]. Riluzole has been demonstrated to alleviate neuropathic pain in DCM rodent model [124]. Pregabalin is a drug commonly used to control chronic neurogenic pain in various conditions. It was found to have a protective effect in OLG from glutamate-induced apoptosis [171]. Other molecules with well-known antioxidant effects like pyrrolidine dithiocarbamate and vitamin E have also shown to have protective effects in OLG against apoptosis [108]. Among them, pregabalin are most well studied in relieving DCM and showed low to moderate evidence for beneficial effects on some neuropathic symptoms [172].

### 8.2. Surgical Treatment

A posterior approach to decompress the spinal canal was the first procedure described in spine surgery. The relative ease of the approach and its reported clinical success made it a common surgery for pathologies such as disc herniations, abscesses and spinal tuberculosis. Eventually, it was used to decompress the cervical spinal cord. Often multilevel, it has shown mixed results over time, and importantly it has been shown to be associated with important complications. Post-laminectomy kyphosis has been described at high rates up to 47% according to some series [173]. The cervical spine transmits close to 1/3 of compressive loads through the vertebral bodies and 2/3 through the posterior elements [174]. Acknowledging this has led to a shift in surgeon´s preference from decompression alone to decompression plus fusion [175]. Recent studies, including a small randomised controlled trial (RCT) have shown that in certain patients, i.e., those with preserved cervical lordosis, decompression alone could be as effective as decompression with fusion [176,177].

A common procedure used to treat DCM is the Anterior Cervical Decompression and Fusion (ACDF) surgery. This procedure has its roots in the realisation by surgeons that disc herniations needed to be removed for the neurological symptoms to improve. Several techniques described the debulking of a herniated disc from a posterior approach but often they would sacrifice nerve roots or require important mobilisation of the spinal cord, which carried severe consequences. With the first anterior approaches described during the second half of the 20th Century, decompression of the cervical spine from the front became a more suitable option and opened possibilities to address issues that were before impossible to take care of like sagittal alignment, cervical spondylosis and segmental instability. The first anterior discectomy/fusion surgeries were described in 1955 and 1958. The first series of cervical arthroplasties was reported in 1966, falling out of favour for some decades until regaining popularity in the 1990’s. Although with modern techniques, the success rate has improved and risks have decreased, some series still report non-union at around 10% and ongoing pain in the same values, if not higher. Another issue with the anterior approach is its ineffectiveness to successfully address multilevel (more than 3) disease in DCM [178]. Advantages of the anterior approach are lower rates of surgical site infection, less postoperative pain and the possibility to address sagittal alignment. The rate of complications such as adjacent segment degeneration and subsidence are still unclear [179].

Laminoplasty is another popular technique, in which the laminae are cut and then moved and fixed in a new position to increase the space of the spinal cord. Proposed advantages of this technique include preservation of the native bone, and movement of the cervical spine and slower progression of myelopathy compared to laminectomy. These advantages, however, have not been shown unequivocally [180,181].

The goal with these procedures is to decompress the encroached spine. However, some issues related with these decompressions have been noticed. Nearly 10% of patients have shown worsening of neurological symptoms and almost half do not show neurologic improvement even six months after the decompression surgery [182]. Ischemia-reperfusion injury (IRI) has been identified as an important mechanism to explain these findings [183]. After blood flow returns to an ischemic spinal cord, a major cytokine release occurs. Cytokines released include TNF-α, CCL-2, CCL-3, CCL-5, CXCL1, IL-1β and IL-6; all of them are associated with a strong local immune response, with the oxidative and apoptotic damage that comes with it [184,185]. Although the exact mechanism remains unclear, several processes play a role, including leucocyte recruitment, cytokine cascades, microvessel endothelial damage and apoptosis [186]. Reperfusion to the site of compression and oxidative damage would explain acute and subacute neurological decline after surgery.

## 9. Future Directions

The lack of diagnostic tools that would enable the detection of DCM from its early stages indicates the need for new research in this area. fMRI and DTI are promising techniques, providing evidence of metabolic changes and microstructural tissue lesions that are impossible to detect with conventional MRI. There is also need for novel imaging techniques, such as diffusion MRI (dMRI), that can provide more information about microstructure. Further research into the molecular mechanisms of DCM is a must. Understanding the mechanisms seen in cervical IVD with those seen in the lumbar spine would be of great value to direct future therapies. The role of previously unsuspected components in the pathophysiology of the disease is just beginning to be elucidated. For instance, the effect of a person’s unique microbiome profile and the inflammatory response it may have locally around the cervical spine and systemically may explain, at least partly, degenerative changes that could lead to DCM. Moreover, as a chronic condition, the profile of DCM biomarkers could help predict flare-ups of the disease, which could assist in choosing therapeutic alternatives better suited to each patient.

## 10. Conclusions

With an aging population, the incidence and prevalence of DCM will continue to increase. The economic burden will soar too, because DCM is a common cause of disability in the aged population. Surgical decompression, although unpredictable, continues to be a common treatment, even though it sometimes leads to worsening of symptoms. The pathophysiology of the disease is not completely understood, and several mechanisms have been postulated to explain it. The key for successfully treating DCM could be partly hidden in the huge array of interactions that take place and have been mentioned in our review. Understanding all the factors associated with this condition will undoubtedly shed some light on future treatment alternatives, not only for this condition, but for many other neurodegenerative conditions that may share similar pathways in their physiopathology.

## Figures and Tables

**Figure 1 jcm-10-01214-f001:**
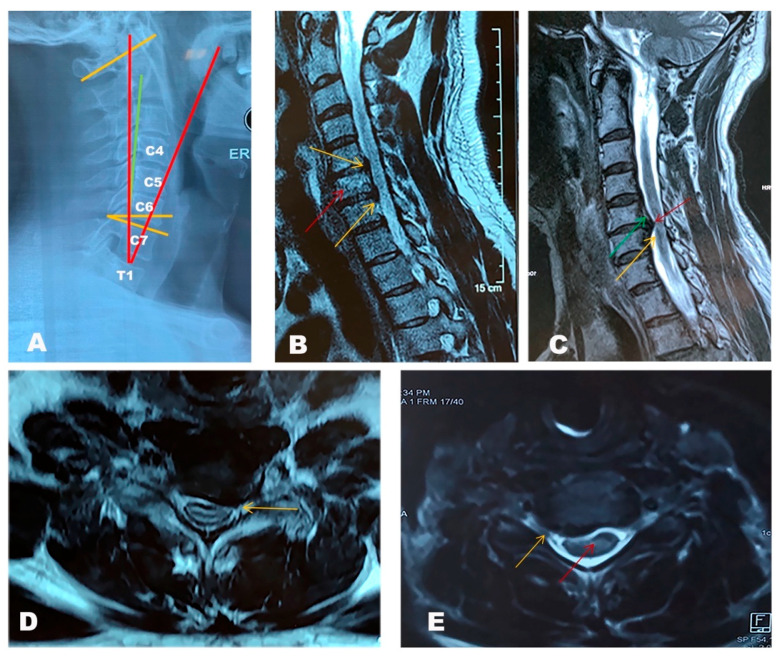
Radiological features of degenerative cervical myelopathy (DCM). (**A**) Standing, lateral X-ray image of a DCM patient showing a normal sagittal balance. In this case, the degeneration did not arise from a severe mal-alignment but rather from degeneration of the structures in the spinal canal. Red: Cervical tilt; Green: cervical sagittal vertical alignment (SVA); Yellow: Cobb angle C1–7 and C7 slope angle. (**B**) T2 weighted sequence of a cervical spine MRI. Sagittal cuts showing C5–C6, C6–C7 and C7–T1 degenerative disc disease with posterior osteophytes compressing the spinal cord at C5–C6 (yellow arrow up) and C6–C7 (yellow arrow down). Type 1 Modic endplate changes at the inferior endplate of C5 and superior endplate of C6 indicate low grade inflammation at this level (red arrow). The relationship between inflammation at the endplates and discs and the presence of bacteria here is unclear. (**C**) Sagittal cuts showing multilevel disc disease with a protruding disc at C5–C6 indenting the spinal cord at this level. Hyper-intensity of the cord can be noticed or a white colour on the cord that under normal circumstances appears as black surrounded by a white signal (the cerebrospinal fluid), demonstrating evidence of myelomalacia (yellow arrow). T2 mapping also showing stenosis of the cervical vertebral canal cause by ossification of the posterior longitudinal ligament (OPLL) (green arrow) with a large osteophyte complex at this level (red arrow). The only symptoms showcased by this patient were mild axial neck pain and bilateral plantar paresthesias. (**D**) Axial cut through the C5–C6 disc showing a left sided disc bulge compressing the exiting nerve root at this level (yellow arrow). (**E**) Axial cut at the C4–C5 level showing a posterior osteophyte complex (yellow arrow) abutting the spinal cord and indenting it. A hyperintense signal can be seen in the cord at this level which could indicate myelomalacia (red arrow).

**Figure 2 jcm-10-01214-f002:**
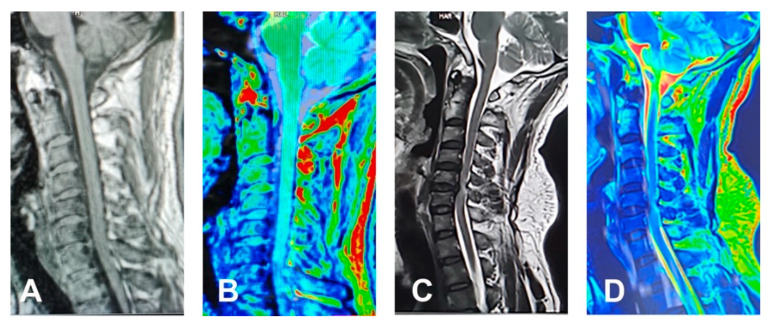
A 38-year-old female presented with history of chronic neck pain: (**A**) No disc herniation and spinal cord compression was showed on sagittal T1 weighted MRI. (**B**) The diffusion tensor imaging (DTI) maps do not show obvious change as well. A 43-year-old female with right brachialgia: (**C**) Sagittal T2 weighted MRI shows spinal cord compression with hyperintense cord signals at C4/5 and C5/6 levels. (**D**) DTI image shows loss of blue colour of the normal cord.

**Figure 3 jcm-10-01214-f003:**
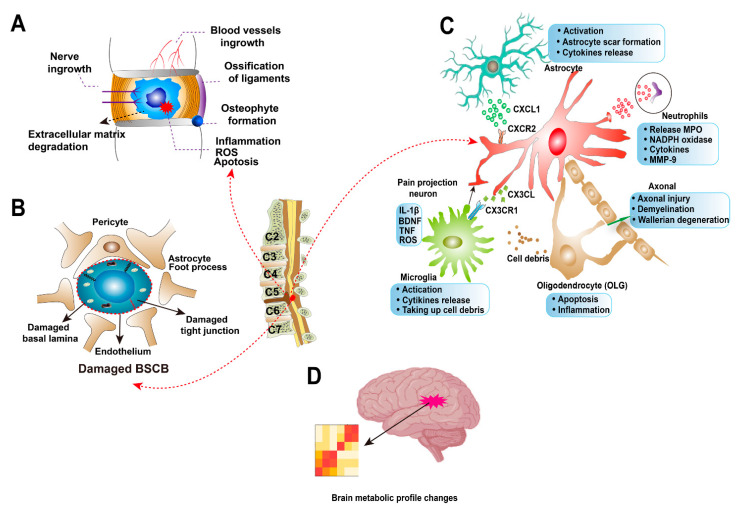
Molecular features of degenerative cervical myelopathy (DCM). (**A**) The hallmarks of cervical disc degeneration. Compared to healthy intervertebral disc, the degenerative disc has increased blood vessel and neuronal ingrowth. Increased inflammation, reactive oxygen species (ROS) and cell apoptosis result in extracellular matrix degradation. The cartilage endplate may be calcified, and osteophytes form on the adjacent vertebral bones. Ossification of the posterior longitudinal ligament (OPLL) can also be found in degenerative cervical spines. (**B**) Blood–spinal cord barrier (BSCB) is disrupted in DCM, with the features of damaged basal lamina and tight junction. (**C**) The roles of cells types in spinal cord during DCM. Astrocyte participates in scar formation in spinal cord; and activated astrocytes can release CXCL1 to interact with CXCR2 receptor on neurons, inducing descending neuron degeneration in spinal cord. CX3CL/CX3CR1 interaction between microglia and neuron regulates neuroinflammation in DCM. Microglia can also take up cell debris from other cells, such as apoptosis oligodendrocytes (OLG). Infiltrating neutrophils release myeloperoxidase (MPO), nicotinamide adenine dinucleotide phosphate oxidase (NADPH oxidase) and other cytokines in the microenvironment. Neutrophils can also express Matrix metalloproteinase (MMP)-9 as a strong pro-inflammatory molecule. (**D**) The brain metabolic profile was found to change in DCM patients.

**Figure 4 jcm-10-01214-f004:**
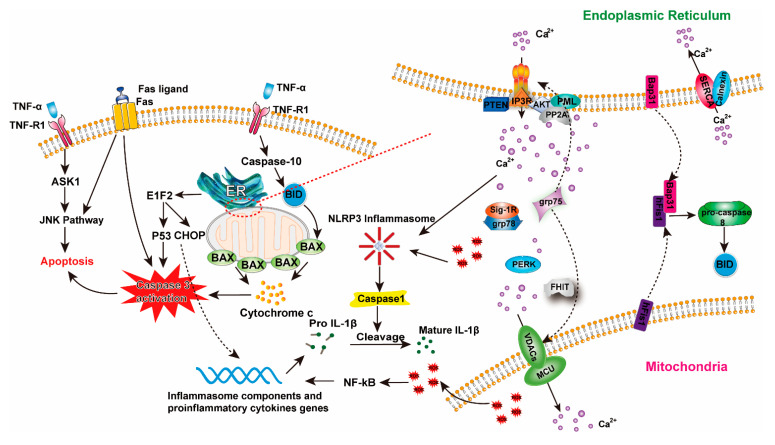
Apoptosis and inflammation regulation in oligodendrocytes (OLG) during DCM. (**Left**) Pathways for apoptosis and inflammation regulation in OLG. In DCM, TNF-α and Fas/FasL pathway and downstream caspase-3 induced apoptosis pathways can be activated. Inflammasome components and proinflammatory cytokines are elevated during the process. (**Right**) Magnification of the ER–mitochondria interactions. Several molecules in contact sites can regulate inflammasome activation, ROS accumulation, Ca^2+^ transfer and apoptosis.

## Data Availability

Not applicable.

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
