# Peer review of "Degenerative Cervical Myelopathy: Insights into Its Pathobiology and Molecular Mechanisms"

_jcm, 2021, doi:10.3390/jcm10061214_

Round 1

Reviewer 1 Report

I suggest to simplify the paragraph 6.2: Genetic Polymorphism. 

The list of the mutations quoted is a little complicated and it is not easy to train the thought. Probably,  the content of this paragraph could be better outlined in a schematic and systematic way with a table. 

Another advise is to add to the references:

-rows 175 and followings: D'Avanzo S et al: The Functional Relevance of Diffusion Tensor Imaging in patients with Degenerative Cervical Myelopathy. J Clin Med 2020 Jun 11; 9(6): 1828

-rows 388 and followings: Colon JM and Miranda JD: Tamoxifen: an FDA approved drug with neuroprotective effects for spinal cord injury recovery. Neurol Regen Res 2016 Aug; 11(8): 1208-1211

Author Response

I suggest to simplify the paragraph 6.2: Genetic Polymorphism.

The list of the mutations quoted is a little complicated and it is not easy to train the thought. Probably, the content of this paragraph could be better outlined in a schematic and systematic way with a table.

Response: Thank you for your valuable suggestion. We have created a table (Table 5) to list the genes associated with DCM pathology.

Another advise is to add to the references:

 -rows 175 and followings: D'Avanzo S et al: The Functional Relevance of Diffusion Tensor Imaging in patients with Degenerative Cervical Myelopathy. J Clin Med 2020 Jun 11; 9(6): 1828

Response: Done.

-rows 388 and followings: Colon JM and Miranda JD: Tamoxifen: an FDA approved drug with neuroprotective effects for spinal cord injury recovery. Neurol Regen Res 2016 Aug; 11(8): 1208-1211

Response: Done

Reviewer 2 Report

Well written, but more description of clinical cervical alignment and Dynamic Factor (Kinematic CT Myelography and MRI) would be helpful.

Author Response

Well written, but more description of clinical cervical alignment and Dynamic Factor (Kinematic CT Myelography and MRI) would be helpful.

Response: Thank you very much for your valuable suggestions. We have added sections for each of these topics in the manuscript.